# Non-hydrostatic effects in the Dead Sea

Oded Padon<sup>1</sup> and Yosef Ashkenazy<sup>1</sup>

<sup>1</sup>Department of Solar Energy and Environmental Physics, BIDR, Ben-Gurion University, Midreshet Ben-Gurion, 84990, Israel

Correspondence to: Y. Ashkenazy (ashkena@bgu.ac.il)

Abstract. The Dead Sea is the saltiest and lowest terminal lake in the world. Currently, the Dead Sea's water level is dropping by more than 1 m per year, due to excessive use of the water that previously flowed into it. The Dead Sea constitutes a unique environment and is important from economic, environmental, and touristic points of view. The winter deep convection of the Dead Sea and its deep and narrow basin suggest that non-hydrostatic effects may significantly affect its circulation. Despite

- 5 these factors, the expected non-hydrostatic effects on the circulation of the Dead Sea have not been investigated. Here we perform high resolution (100 m) ocean general circulation model (the MITgcm) simulations of the Dead Sea and show that the non-hydrostatic results are very different from the hydrostatic ones. Specifically, we show that the winter non-hydrostatic simulations resulted in a layer of dense water overlaying slightly lighter water during the several last hours of the night; this convection process involved plumes of heavier sinking water and the entrainment of the plumes. We also studied the effect
- of the wind stress's diurnal variability and found it to be important, especially during the summer when the wind's variability 10 drastically increased the surface kinetic energy; however, it did not alter the depth density profile. The results presented here may be important for the Dead Sea's potash industry and for the planned Red Sea-Dead Sea canal that aims to stop and, possibly, to increase the level of the Dead Sea using the Red Sea's water.

## 1 Introduction

The Dead Sea is the saltiest and lowest natural water body in the world (Hall, 1978). It is an elongated (on the south-north 15 axis) terminal lake located between Israel and Jordan, along the Great Syria-African Rift. The Dead Sea surface is currently  $\sim$ 430 m below mean sea level, and its deepest point is  $\sim$ 730 m below mean sea level, such that its maximal depth is currently  $\sim$ 300 m. The current dimensions of the Dead Sea are approximately  $20 \,\mathrm{km} \times 50 \,\mathrm{km}$ . The Dead Sea is located in an extremely arid region with an average precipitation rate of  $\sim 5 \,\mathrm{cm}\,\mathrm{vr}^{-1}$ . Its main water source is the Jordan River, whose current runoff is equivalent to  $\sim 10 \,\mathrm{cm}\,\mathrm{yr}^{-1}$  (Lensky et al., 2005). 20

The salinity of the Dead Sea's water is about eight times larger than standard ocean water salinity (Steinhorn, 1985, 1991; Gertman and Hecht, 2002), reaching values of  $\sim$ 280 ppt. This high salinity is due to the fact that the Dead Sea is a closed basin without outlets, so that natural water loss from the Dead Sea only occurs via evaporation or human usage (such as water pumping for the potash industry). The current evaporation rate exceeds the precipitation and river runoff rates, such that the

25 Dead Sea's salinity increases with time, and the sea surface level decreases with time by more than 1 m per year (Hecht et al.,

1992; Yechieli et al., 1998; Gertman and Hecht, 2002). This also leads to what is known as salt precipitation in the deep water, with salt accumulation at the bottom of the Dead Sea occurring at a rate of  $\sim 10 \,\mathrm{cm}\,\mathrm{yr}^{-1}$  (Lensky et al., 2005).

For the last several centuries until the mid-1970s, the Dead Sea was stratified throughout the year, in a state known as meromictic stratification (Neev and Emery, 1967; Anati, 1997). During winters, the surface water became colder, but since it

5 was relatively fresh, it was not dense enough to sink to the bottom of the lake. In the 1970s, the Dead Sea experienced a drastic change in its circulation, as the meromictic stratification was replaced by total vertical mixing during the winter (Steinhorn and Gat, 1983; Steinhorn, 1985; Anati et al., 1987; Gertman and Hecht, 2002; Steinhorn et al., 1979), thus switching to its current holomictic state.

Increasing human use of Lake Kinneret waters (the Sea of Galilee, located about 100 km north of the Dead Sea, along the
African-Syrian Rift) has decreased the Jordan River runoff to the Dead Sea, making its surface water saltier, and thus reducing the salinity gradient between the surface water and the deep water. Moreover, water from the Dead Sea's main northern part is pumped to the sea's former southern part that now contains evaporation ponds used for the potash industry. During cold winter nights, the surface water becomes colder, and since the surface layer has become saltier over the years, the surface water is now heavier than the deep water, leading to deep vertical mixing. Nowadays, except for in years with exceptionally
high precipitation (such as the winter of 1991-1992), the Dead Sea's water column experiences total mixing every winter. This

15 high precipitation (such as the winter of 1991-1992), the Dead Sea's water column experiences total mixing every winter. This process of vertical mixing by the sinking of cold surface water is associated with non-hydrostatic dynamics.

The Dead Sea is currently of local and global interest, due both to the economic importance of the Dead Sea Works and to tourism associated with the Dead Sea, as well as to the planned Red Sea-Dead Sea conduit. This conduit (Beyth, 2007; The World Bank, 2013) aims to transport water from the Gulf of Eilat/Aqaba to the Dead Sea, in order to stop the drop in

- 20 the Dead Sea's level (and possibly restoring it to its higher 1970s level), and to use the ~430 m elevation difference between the Gulf of Eilat and the Dead Sea as an energy source for water desalination. The ocean water imported through the planned conduit is expected to have a significant impact on the conditions of the Dead Sea (Asmar and Ergenzinger, 2002, 2003). More particularly, the imported water may affect the overall circulation, the surface water conditions, the water chemistry and the ecological system surrounding the Dead Sea, and may have severe implications for the Dead Sea's environment and shores.
- 25 Thus, understanding the dynamics of the Dead Sea and predicting the effects of the Red Sea-Dead Sea conduit are of major importance from both local and global perspectives.

The Dead Sea's circulation has been studied using both simple models (Vadasz et al., 1983; Asmar and Ergenzinger, 2002, 2003) and more complex general circulation models (GCMs) (Ezer, 1984; Sirkes, 1986), and has been simulated mainly using 1D and 2D models. Modeling the impact of the high density of the Dead Sea water on the circulation show that the near-surface

- 30 wind-driven currents in the Dead Sea are ~15% weaker than currents in standard seawater under the action of the same wind (Huss et al., 1986). These simulations were not long enough to study the dynamics incurred by the annual cycle and were too coarse to study the non-hydrostatic effects of the circulation. The Dead Sea's wind-driven seiches were studied by Sirkes (1986) using a seiche model and a simple oceanic CGM, under the assumptions of homogeneous temperature and salinity fields. To our knowledge, the Dead Sea's circulation has not been studied using a non-hydrostatic GCM, even though non-hydrostatic
- 35 effects are expected to be significant in this circulation, due to winter mixing (deep convection) of the water column and due

to the deep and narrow shape of Dead Sea. The goal of the present study is to investigate whether and how non-hydrostatic effects impact the Dead Sea's properties and circulation. Using a high resolution oceanic GCM, the MITgcm (MITgcm-group, 2010), we show that the non-hydrostatic dynamics resulted in a Dead Sea circulation that is much different than the hydrostatic one, suggesting that non-hydrostatic effects must be taken into account in order to more accurately simulate the Dead Sea's

5 circulation-accurate simulations of the lake's circulation and salinity distribution are important for optimal management of the potash industry and for the planned Red-Sea-Dead-Sea canal. Previous studies had used the MITgcm (and other GCMs) to compare between hydrostatic and non-hydrostatic simulations (e.g. Haine and Williams, 2002; Legg et al., 2006; G. Sannino et al., 2014; Magaldi and Haine, 2015; McKiver et al., 2016).

In this paper, we first describe the model, setups, and the experiments (Sec. 2). We then present the results for annual cycle 10 forcing (Sec. 3), for the winter diurnal cycle (Sec. 4), and for summer and winter diurnal cycles under fixed and diurnally varying winds (Sec. 5). We then summarize and conclude the paper (Sec. 6).

## 2 Model Setup

## 2.1 The MITgcm

The Massachusetts Institute of Technology general circulation model (MITgcm) (MITgcm-group, 2010) is a widely used, open source GCM, capable of simulating the dynamics of both the atmosphere and the ocean, separately or in a coupled mode (Marshall et al., 1997a; Adcroft et al., 2002). It is a *z*-coordinate model, and its main properties are: (i) hydrostatic and full non-hydrostatic (and non-Boussinesq) capabilities, (ii) a finite volume numerical model with partial cells, and (iii) adjoint modeling. In particular, the support for non-hydrostatic dynamics allows the model to be used to study both large-scale and global dynamics, and small-scale processes. It also has many parameterizations to account for various oceanic processes.

- 20 Non-hydrostatic simulations are numerically more complicated (and usually much more computationally "expensive") compare to hydrostatic simulations. In the non-hydrostatic option of the MITgcm a 3D elliptic equation for the pressure is solved every model's time step when Neumann boundary conditions are used. However, in hydrostatic simulations a 2D elliptic equation is solved every time step, making its the computational cost much cheaper compare to non-hydrostatic simulation. When the non-hydrostatic effect are absent or insignificant the 3D elliptic solver converges very quickly and the computation time is similar to the computation time of hydrostatic simulation (MITgcm-group, 2010). The computational cost of non-hydrostatic
- simulation increases as the non-hydrostatic effect becomes more significant.

#### 2.2 Setup

30

To simulate the Dead Sea's circulation, we used a cascade of three computational grids, with horizontal grid resolutions of 400 m, 200 m and 100 m, as detailed in Table 1. We used 15 levels in the vertical direction with variable resolution as follows (top to bottom): 5, 5, 5, 5, 5, 7, 10, 15, 20, 25, 30, 35, 40, 45, and 52 m; the spacing between the vertical levels was chosen to allow a better simulation of dynamics near the surface, which are affected by temperature and salinity forcing and by the wind

5

10

stress. [We have repeated the simulations of some of the hydrostatic and non-hydrostatic runs using 100 vertical levels (with 1 m resolution at the surface layers) and obtained qualitatively similar results.] The 400 m simulation was started with the water at rest and with uniform temperature and salinity fields matching the surface conditions, roughly mimicking winter conditions. This simulation was run under annual cycle (monthly mean) forcing of surface temperature and salinity and constant northerly winds: it was run for twenty years to a steady state, representing the Dead Sea's circulation under annual cycle forcing.

The state of the 400 m simulation after twenty years was used as the initial condition for the 200 m simulation (using spatial interpolation). This simulation was run with the same annual cycle and constant winds as the 400 m simulation and for three years, until reaching a steady state, representing a higher resolution annual cycle circulation. The states of the 200 m simulation in both summer and winter were then used as initial conditions for the 100 m simulations, which were run with a diurnal cycle in temperature, constant salinity matching the season, and both constant and varying winds.

The described process of cascading from the 400 m to the 100 m simulation was repeated using both hydrostatic and non-hydrostatic modes, to compare between the two. In all hydrostatic simulations and the low resolution non-hydrostatic simulations (400 m and 200 m), implicit vertical diffusion was used; the implicit vertical diffusion scheme is a standard vertical mixing scheme (MITgcm-group, 2010) in which the vertical diffusion is drastically increased (i.e., from a value of  $10^{-5} \text{ m}^2 \text{ s}^{-1}$ 

to  $10 \text{ m}^2 \text{ s}^{-1}$ ) when the water column becomes unstable. To validate our results we have re-run some of the high resolution hydrostatic simulations using the KPP scheme (Large et al., 1994; Mellor and Yamada, 1982; Ezer, 2005) and obtained similar results. In the 100 m non-hydrostatic simulations, implicit vertical diffusion was not used, as these simulations have a high enough resolution to simulate vertical convection without parameterization.

Where used, the implicit vertical diffusion coefficient was 10 m<sup>2</sup> s<sup>-1</sup>; we also examined the value of 1 m<sup>2</sup> s<sup>-1</sup> and obtained
very similar results. Under regular stable conditions, the vertical diffusion coefficient of temperature and salinity was chosen to be 10<sup>-5</sup> m<sup>2</sup> s<sup>-1</sup>, which is a typical value for open ocean eddy parameterized diffusion coefficient (Wunsch and Ferrari, 2004). The horizontal diffusion coefficient was chosen according to the horizontal grid resolution and was ranging from 4 m<sup>2</sup> s<sup>-1</sup> for the 400 m horizontal resolution runs to 1 m<sup>2</sup> s<sup>-1</sup> for the 100 m horizontal resolution runs. The vertical viscosity coefficient was chosen to be 5 × 10<sup>-4</sup> m<sup>2</sup> s<sup>-1</sup> and the horizontal viscosity coefficient varied according to the horizontal grid resolution, from 4 m<sup>2</sup> s<sup>-1</sup> for the 400 m horizontal resolution runs to 1 m<sup>2</sup> s<sup>-1</sup> for the 100 m horizontal resolution runs.

As described above, the numerical simulations involve the solution of 2D and 3D elliptic equations, at every time step-these are solved iteratively until reaching a maximum number of iterations or until the residual value drops below a target value. In all simulations, the 2D solver target residual was  $10^{-13}$ , and in the non-hydrostatic simulations, the 3D solver target residual was  $10^{-9}$ . Both solvers were allowed enough iterations to reach the target residual. The time step used was adjusted to the

30

horizontal resolution, and is listed in Table 1; in all runs the time step was small enough to satisfy the CFL condition and the Courant number was usually much smaller than 1. Realistic, high resolution Dead Sea bathymetry (Fig. 1) was used, based on Hall (1978).

In the simulations presented below we have used the nonlinear free surface option of MITgcm. In addition, we used the nonlinear flux limiter advection scheme (advection scheme 33) of MITgcm.

Standard equations of state are invalid for the Dead Sea due to its extremely high salinity. Instead, we used a linear equation of state, based on Steinhorn (1991) and Ezer (1984):

$$\rho = \rho_0 \left[ 1 - \alpha (T - T_0) + \beta (S - S_0) \right] \tag{1}$$

where ρ<sub>0</sub> = 1237 kg m<sup>-3</sup>, α = 3.5 · 10<sup>-4</sup> C<sup>-1</sup>, β = 9.5 · 10<sup>-4</sup>, T<sub>0</sub> = 23 °C, and S<sub>0</sub> = 278 ppt. We note that the thermal and
haline coefficients, α, β, are different than the ones used in Steinhorn (1991) and Ezer (1984), to match the saltier present day water of the Dead Sea. Since the chemical properties of the Dead Sea water are evolving with time, the linear equation of state (1) only crudely determine the density.

In all simulations, surface forcing took the form of temperature and salinity restoring, and prescribed wind stress. We use restoring boundary conditions instead of surface fluxes as the former can be estimated more easily from observations; we note

10 that the use of heat and freshwater surface fluxes often yields more realistic mixed layers (e.g., Ezer, 2000). The forcing salinity and temperature used was a sine wave, given by:

$$T(t) = \frac{T_{\min} + T_{\max}}{2} - \frac{T_{\max} - T_{\min}}{2} \cos\left(2\pi \frac{t}{t_{\text{cycle}}}\right)$$
(2)

$$S(t) = \frac{S_{\min} + S_{\max}}{2} - \frac{S_{\max} - S_{\min}}{2} \cos\left(2\pi \frac{t}{t_{\text{cycle}}}\right),\tag{3}$$

where t<sub>cycle</sub> is one year in the annual cycle simulations, and one day in the diurnal cycle simulations. The values of T<sub>min</sub>, T<sub>max</sub>,
S<sub>min</sub>, S<sub>max</sub> are based on measurements performed by the IOLR (Israel Oceanographic and Limnological Research) (IOLR, ISAMAR, 2009), and are listed in Table 1. The restoring times of the forcing salinity, S(t), and temperature, T(t), were 12 days for surface salinity and 4 days for surface temperature.

The wind stress used in the constant wind simulations matches a northerly wind of  $4.5 \,\mathrm{m\,s^{-1}}$ , which is a typical wind for the Dead Sea area. Based on Gill (1982), we applied the following conversion from wind velocity,  $\mathbf{v}_{air}$ , to wind stress,  $\tau$ :

20 
$$\tau = \rho_{\text{air}} C_D \mathbf{v}_{\text{air}} |\mathbf{v}_{air}|,$$
 (4)

where  $C_D = 10^{-3}$  when  $|\mathbf{v}_{air}| 

5

both in hydrostatic and non-hydrostatic modes. Fig. 3 depicts the sea surface temperature (SST) and velocity field snapshots from the 200 m simulation at the end of the three-year runs, in summer and winter, for both the hydrostatic and non-hydrostatic simulations. As this figure shows, the hydrostatic and non-hydrostatic results are quite similar with a greater difference during the winter. The similarity between the hydrostatic and non-hydrostatic simulations is expected, as spatial resolutions of 400 and 200 m might be too coarse to make the non-hydrostatic effects significant.

Fig. 4 shows the density, salinity, and temperature fields of the 200 m non-hydrostatic run, averaged over the entire lake in the horizontal directions, presented in the depth-time plane. The results of the hydrostatic simulations are almost identical to the non-hydrostatic ones. The profiles are uniform below a certain mixing layer. The depth of the mixed layer varies between 20 m in summer to complete mixing of the water column. The surface layer becomes deeper as the surface water becomes colder and thus denser until at some point during the winter, the mixing layer disappears and total mixing occurs. This state

10 colder and thus denser until at some point during the winter, the mixing layer disappears and total mixing occurs. This state persists for approximately two months in the winter, before the mixing layer forms again as the surface water becomes warmer and lighter. The results described above resembles observations from the Dead Sea reported in IOLR web-page, and in other studies (e.g., Anati, 1997; Gertman and Hecht, 2002; Gertman, 2012).

A useful measure of the flow is the stream function,  $\psi$ . We calculated the stream function (that reflects the overturning 15 circulation) in the Y-Z plane

$$\psi(y,z) = \int_{-H}^{\tilde{\nu}} V(y,z') dz', \tag{5}$$

where  $V = \int v \, dx$  (i.e., integration over the entire width of the lake). The stream function,  $\psi$ , indicates the net northward volume flux through the X-Z plane located at y and stretching from the bottom up to z.

- The overturning circulation in the Dead Sea peaks at February and then drastically relaxed during March. In Fig. 5 we present typical snapshots of the stream function from the 200 m simulations, of both the hydrostatic and non-hydrostatic runs during these two months. The February snapshots show a single circulation cell with a counterclockwise flow, occupying the entire lake. The southward flow near the surface corresponds to the action of the northerly winds. The March results show the counterclockwise cell compressed to the top 20 m near the surface, with a clockwise circulation cell occupying the deep water. Note that as seen in Fig. 4, in February, the entire water column is mixed, while in March, the mixing layer is
- 25 forming, occupying approximately the top 20 m near the surface. The transport during February is about five times larger than during March. By comparing the hydrostatic and non-hydrostatic stream function results, we see that they have non-negligible differences, but that they share the main features of the circulation. We also obtained similar stream functions for the hydrostatic and non-hydrostatic runs for the different months throughout the year. We thus conclude that there are no significant differences between the hydrostatic and non-hydrostatic simulations for the 200 m resolution case, either in the Dead Sea's circulation or
- 30 its properties (i.e., temperature, salinity, and hence density). Therefore, we proceeded to study possible non-hydrostatic effects in a finer resolution setup when the diurnal cycle is taken into account.

5

#### 4 Results of Winter Diurnal Cycle Forcing

As described in Sec. 2.2, the final states of the 200 m simulations in both winter and summer were used as input for the 100 m simulations with a diurnal cycle in winter and summer, and with both constant and time varying wind stress. In this section, we focus on the diurnal cycle in winter, with constant winds, and compare the hydrostatic results to the non-hydrostatic ones. We expected to see a marked difference due to the presence of significant vertical convection: during cold winter nights, the surface water becomes colder and denser than the deep water, which results in convective instability, leading to non-hydrostatic

vertical convection. A spatial resolution of 100 m should suffice to simulate such convection events (Marshall et al., 1997b).

The diurnal cycle simulations were run for sixty days, and we present the results of the last five days of the simulation. Sixty days proved to be enough to switch from the 200 m resolution to the 100 m resolution and from the annual cycle to the diurnal

10 cycle. We note that in contrast to the annual cycle simulations with the seasonal diurnal cycle, it did not make sense to run the simulation until it reached a steady state, as a perpetual January or a perpetual July until a steady state is reached is not representative of actual circulations (as opposed to a perpetual annual cycle until reaching steady state), which are (seasonally) transient by their nature.

The horizontal mean versus depth and time of the density, salinity, and temperature of the hydrostatic and non-hydrostatic runs are presented in Fig. 6. Only the top 100 m of the water column is shown as below this depth, the water properties are almost uniform. Both figures clearly present the effect of the diurnal cycle on the top water layer near the surface, which becomes colder and denser during the nights, and warmer and lighter during the days.

While both the hydrostatic and the non-hydrostatic simulations presented the effect of the diurnal cycle, there was a significant difference between them. In the hydrostatic results (Fig. 6), the surface water got colder and denser during the night,

- 20 but it was almost never denser than the water below the surface, since the surface water immediately mixed once it became denser than the water below it. In the non-hydrostatic run (right panels of Fig. 6), the surface water did become colder and denser than the water below it during the night, but this state persisted until the morning. Moreover, the nocturnal SST of the non-hydrostatic run was much colder than that of the hydrostatic run as a result of its slower sinking. However, neither the hydrostatic nor the non-hydrostatic simulations presented in Fig. 6 shows complete mixing during the night as we would expect, but this is a result of the horizontal averaging over the entire lake when, in fact, the density (depth) profile varies from
- one location to another. As we shall show below, the actual density profiles do present total mixing during the night.

The left two panels of Fig. 7 shows the SST and velocity field snapshots at 4 am, for both the hydrostatic and non-hydrostatic simulations—the hydrostatic results are significantly different from the non-hydrostatic ones. The surface water was colder in the non-hydrostatic simulation, as already indicated in Fig. 6, and the spatial features of the velocity field also had non-neglicible differences.

30 negligible differences.

As suggested by Fig. 6, the most apparent difference between the hydrostatic and non-hydrostatic simulations was the different dynamics of the surface water density relative to the deep water density. For this reason, we studied the density difference between the surface (upper layer) and deep water, taken as the mean density below 100 m. The right two panels of Fig. 7 shows a snapshot of this density difference at 4 am (approximately the time of minimum SST), of both the hydrostatic and

5

non-hydrostatic simulations, with the contour of  $\Delta \rho = 0$  overlayed. This figure demonstrates a significant difference between the hydrostatic and non-hydrostatic simulations. As seen, in the non-hydrostatic simulation, the northern three-quarters of the lake's surface water was denser than the deep water by about  $0.05 \,\mathrm{kgm^{-3}}$ , equivalent to a relative density difference of  $\Delta \rho / \rho \sim 3.9 \times 10^{-5}$ . However, for the hydrostatic case, the surface water was almost everywhere lighter than the deep water, except for a very small region in the northeast. Even there, the density difference was less than  $0.001 \,\mathrm{kgm^{-3}}$ , i.e., a relative difference of  $\Delta \rho / \rho 

The major discrepancy between the hydrostatic and non-hydrostatic runs, presented in Figs. 6–10, can be explained by the different ways that the vertical convection was simulated in the two modes. In the hydrostatic simulations, we used the implicit vertical diffusion scheme when the water column destabilized; this caused any unstable vertical density gradients to diffuse and disappear rapidly. Essentially, this means that the hydrostatic simulation did not allow the surface water to become denser than

- 5 the water below the surface, which led to the cut-off that appears in Fig. 10. This is the reason that in the snapshot presented in the two right panels of Fig. 7, the hydrostatic simulation has almost no area in which the surface water is denser than the deep water. However, in the non-hydrostatic simulation, implicit vertical diffusion was not used, and the physical process of cold and dense surface water sinking by vertical convection was fully simulated, resulting in plumes such as those presented in Fig. 8. This means that the non-hydrostatic simulation did sustain the unstable state of heavy water above lighter water for
- 10 several hours of surface water cooling during the night during which plumes of downward and upward moving water occurred. This situation was probably related to the fact that the SST continued to drop during the night, creating increasingly heavier water that maintained the unstable state of heavier water above lighter water throughout the night. A situation similar to the non-hydrostatic results presented above has been observed in the Gulf of Elat (see Fig. 2 of Biton et al., 2008) where dense water was observed to overlay lighter water. The fact that such a seemingly unstable situation has been observed in a nearby
- 15 environment supports the conclusion that the non-hydrostatic simulation provided a better description of the actual physical process, which as explained above, allows the seemingly unstable situation to exist. Such conditions may be important for the potash industry, to optimize their production. A similar situation was observed in the Dead Sea measurements performed by the IOLR (on their online near-real-time monitoring web-page) e.g., during 12-14 of March 2014 and 26-28 Feb. 2015 where cold surface water overlaid warmer water for more than 4 hours; see Fig. 11.
- As described in Sec. 2.2, the implicit vertical diffusion coefficient used was  $10 \text{ m}^2 \text{ s}^{-1}$ . We also repeated the diurnal cycle winter experiment using a coefficient of  $1 \text{ m}^2 \text{ s}^{-1}$ , but this did not change any of the results presented above. The hydrostatic simulation still did not allow heavy water above light water, and it imposed the non-physical cut-off on the surface water density.

The difference in the way the hydrostatic and non-hydrostatic modes simulated the vertical mixing has a significant impact on the overall circulation of the Dead Sea. Fig. 12 presents vertical profiles of density, salinity and temperature, horizontally averaged over the entire lake and time averaged over the last five days of the simulation (days 55-60). As seen, in the hydrostatic results, the deep water had a lower temperature, slightly lower salinity, and, overall, a higher density than the deep water in the non-hydrostatic simulation. This is explained by the fact that the diffusive mixing was stronger than the actual, convective mixing, which made the deep water respond more strongly to surface conditions in the hydrostatic simulation than

30 in the non-hydrostatic one. Thus, in winter, when the overall lake becomes colder and fresher, these changes affected the deep water more in the hydrostatic simulation than in the non-hydrostatic simulation.

5

# 5 Results of the Summer and Winter Diurnal Cycle Under Fixed Diurnally Varying Winds

After studying the hydrostatic and non-hydrostatic effects under the action of temporally constant winds on the Dead Sea's dynamics and properties, we then focused on more realistic non-hydrostatic configuration and studied the effect of wind (temporal) variability on the sea's dynamics; we note that the winds in the simulations are spatially uniform. As described in Sec. 2.2, the time varying winds were based on the climatological means of observed winds in the Israel Meteorological

Service's Ein Gedi station (presented in Fig. 2).

The surface snapshots of temperature and velocity field spanning the diurnal cycle at six-hour intervals are shown in Fig. 13. In particular, this figure shows that the velocity field of the two runs differed greatly, as one would expect, as the wind stress significantly affected the surface current field. Similar difference between constant and time varying wind stress is also found

- 10 in the mean fields of density, salinity, temperature, and velocity (not shown). Still, in both runs, during winter, the northern part of the lake was denser, saltier, and warmer than the southern part. This difference between the northern and southern parts of the lake is partially related to the shallower depths of the southern parts that lead to more effective nocturnal cooling (and more effective relaxation to the fresher winter surface conditions) and in part to the northerly winds that upwell deep dense water in the northern part of the basin.
- 15 The mean density and square of the current speed of the surface water, for the non-hydrostatic winter simulations with both constant and varying winds, are shown in the upper two rows of Fig. 14. The mean current speed squared represents the kinetic energy of the water layer near the surface, which is mainly driven by the winds. As shown, the simulations with constant winds exhibited smaller temporal variability of kinetic energy compared to the time varying winds, especially during the summer. The differences in the surface density were very small.
- In addition to the above, it is noticeable that there is a connection between the surface water density and the surface current kinetic energy. During the day, when the surface water was lighter than the water below the surface, the water column was stratified, and, in practice, the wind forced a thin surface layer of water, which resulted in higher velocities of the surface water. During the late night and early morning, total mixing occurred, and the mixing acted to decrease the velocity field, as the wind forced a deeper layer of water. We also observed a "bump" in the velocity squared in both simulations, which occurred simultaneously with the density maximizing, which is when vertical convective mixing was at its peak.

We now present the results of the simulations of the diurnal cycle in the summer. The conclusions of the comparison between the hydrostatic and non-hydrostatic cases in the winter diurnal cycle are also valid for the summer simulations, although the differences are less dramatic in the summer. Still, the non-hydrostatic mode is more suitable for simulating the circulation, as vertical (night) convection is a significant process even in the summer. Thus, we present the results from the non-hydrostatic

30 simulations with a diurnal cycle in summer, for both constant and time varying winds. As seen in Fig. 2, the observed winds in the summer presented a high peak of westerly wind during the evening, centered around 7 pm; probably, this peak in wind activity was associated with the Mediterranean Sea breeze, which reaches the Dead Sea only during the late afternoon hours. We expected this peak of strong westerly wind to affect the circulation.

Snapshots of surface temperature and velocity field spanning the diurnal cycle of the summer (July) simulations are shown in bottom two rows of Fig. 13. This figure as well as mean surface density, salinity, temperature, and velocity (not shown) indicate that in the summer, the winds had a great impact on the surface conditions, where the temperature and velocity fields of the two cases were very different.

- 5 The two lower rows of Fig. 14 present the mean density and current speed squared (representing the kinetic energy) of the surface water, from the non-hydrostatic summer simulations. The current speed squared curves show a very dramatic difference between the time varying and constant wind simulations, which is due to the large difference between the varying wind stress (see Fig. 2) and the constant northerly winds used in the latter simulation. Specifically, the kinetic energy of the time varying wind was much higher than the one under the action of constant winds, where the latter was almost constant with time and had
- 10 the minimum value of the time varying kinetic energy. However, in spite of the dramatic difference in the kinetic energy of the surface water, the density profiles show very similar behavior, with only a slight offset between the constant wind and the varying wind simulations, where the surface water density was lower in the varying wind simulation.

#### 6 Summary and Conclusions

The Dead Sea is a lake with the saltiest natural water in the world, with a salinity that is about eight times greater than standard
ocean water (Gertman and Hecht, 2002). During winter, the Dead Sea's circulation is significantly affected by the sinking of
cold surface water, leading to a total mixing of the water column. This process is linked to non-hydrostatic processes. The
Dead Sea is currently of local and global interest, as the planned Red Sea-Dead Sea canal (Beyth, 2007) may significantly alter
the circulation in the Dead Sea, which may, in turn, cause negative economic impacts on the Dead Sea Works and Dead Sea
tourism. Despite the clear importance of the Dead Sea and despite the expected non-hydrostatic effects on the sea's circulation
and properties, no studies have investigated the Dead Sea using non-hydrostatic equations. In addition, studies of the effect of

diurnal wind variability under non-hydrostatic effects are also lacking. In this paper, we aimed to begin filling these gaps.

We presented numerical (MITgcm) simulations of the Dead Sea's circulation, using both the hydrostatic and the nonhydrostatic modes. The results demonstrated that for simulations with a spatial resolution of 200 m, the hydrostatic and nonhydrostatic modes are similar. However, in simulations with a spatial resolution of 100 m, the non-hydrostatic mode yielded

25 very different results compared to the hydrostatic mode, especially during winter. During winter nights, the surface water layer cooled, and the density of the surface water became slightly higher than that of the deep water. This instability led to sinking and to a total mixing of the water column in the morning.

The results show that the daily and seasonal mixed layer are dominated by surface cooling and convection with lesser role for wind-driven mixing; this is with the exception an episodic strong wind. The results contrast with the more regular cycles of wind-driven mixed layer dynamics in most oceans (see, e.g., Ezer, 2000). As discussed in Sec. 4, in the 100 m winter simulations, during winter nights, the results from the hydrostatic and the non-hydrostatic modes deviated significantly. In the hydrostatic mode, in practice, the surface water did not become denser than the water below the surface, as the water

column was quickly mixed due to the implicit vertical diffusion. In the non-hydrostatic fine resolution simulations, the vertical