# Peer review of "Non-hydrostatic effects in the Dead Sea"

_Ocean Science, 2017_

## Referee Comment (RC1) · T. Ezer (Referee) · 30 Jun 2017

Numerical modeling of the circulation and mixing processes in the extreme environment of the Dead Sea (DS) is a very challenging task, so progress has been very limited since the development of the first coarse-resolution circulation models of this lake more than three decades ago (Ezer, 1984; Sirkes, 1986). Therefore, the current study, using advanced high-resolution, non-hydrostatic MITgcm, provides new insights into the dynamics of the DS, with focus on non-hydrostatic processes. Because of the unique characteristics of the DS, adapting to the DS model schemes and parameterizations derived for other oceans is usually not appropriate. Therefore, even this advanced model lacks some realism in aspects such as the equation of state, wind drag coefficient, lack of freshwater input and possibly unrealistic mixing schemes. Never-

theless, in my opinion, in the context of model sensitivity experiments to compare hydrostatic versus non-hydrostatic dynamics, the study is important and novel enough to be published, if these limitations are acknowledged. Since the focus of this study is on the surface mixing process, clarifications are needed about the (somewhat unrealistic) model mixing parameters and imposed surface boundary conditions. It seems that the main difference between the hydrostatic and non-hydrostatic simulations during winter nights is a direct result of the different vertical mixing used in those cases and the question is whether this is an artifact of the particular mixing used by the model or a more general result. The paper is generally quite well written (though some text and figures can be improved; see comments below) and the interesting results are clearly presented.

Major comments:

1. The main concern is the vertical mixing used, which needs further clarifications (p. 4). In the hydrostatic model runs, slight instability due to surface cooling caused immediate strong mixing- this is typical for mixing that depends say on the Richardson number or based on a stability function in a turbulence scheme like the Mellor-Yamada (M-Y) model, but here it seems that (inappropriate) constant vertical diffusion coefficient is used everywhere, which do not explain the results. Line 15 is confusing, since only Large et al. (1994) describe the KPP, while Mellor and Yamada (1982) and Ezer (2005) describe and use the M-Y turbulence scheme, not KPP. The imposed restoring surface BCs (p. 5) can really limits surface variability and strongly impact the mixed layer variability (especially for hydrostatic models), so one wonders how different the results would have been without this restriction. Also, fresh water input from the Jordan river and winter flash floods which are neglected here may play a major role in the seasonal changes in stratification – this is only acknowledged at the end of the paper, but this limitation on the salinity and density fields should be clearly indicated when the model setup and surface BCs are described.

2. Though this is a numerical modeling study, it would be useful to add (if possible,

given the limited availability) some direct comparison with observation, to show at least that the seasonal variations in temperature profiles are reasonable (measurements by Anati, Hecht, Gertman and others are cited, and a 1997's book on the DS has further data). The only showing of some observations (Fig. 11) is difficult to relate to any model shown results (is it possible to show example of similar model results?).

Other comments:

3. While the paper is quite well written, the text can be improved- some examples: - The second sentence of the abstract is awkwardly written - the water level of the DS "has been dropping" for a long time, it is not a new phenomenon, as can be interpreted here. - P. 2, line 14, "denser" instead of "heavier". - P. 10, title- "Diurnal Cycle Under Fixed Diurnally Varying Winds" is a confusing statement (is it fixed or varying?), what is probably meant is spatially even but temporally varying. And same page line 5- what is meant by "climatological means"? hourly, daily, monthly means?

4. Figures can be improved with clarifications in captions- some suggestions: - Fig. 1- since it is based on data from 1978 when water level was much higher, it should be clarified for what year the shown shoreline is and what is the water level at that year. - Fig. 2- may be show wind vectors instead of components?, also caption should state if these data are based on hourly observations and what source. - Fig. 4- may be show only the top 100 m (like other figures) to see the details.

5. P.2, near bottom, to first paragraph of p.3- "non-hydrostatic effects are expected to be significant"- is it possible to quantify this assumption based on say scaling arguments such as ratio of horizontal and vertical velocity scales in the DS vs open oceans and the very steep bottom slopes of the DS?. Also, when citing past studies of hydrostatic vs. non-hydrostatic comparisons, one should add that these studies may not be applicable to the unique environment of the DS. For example, in the DS the very high salinity and variations in salinity (which are ignored here) play a major role in the static stability compared to the major role of temperature in most other oceans.
6. P.3- in the description of the model, it is not clear if some of the features are actually used here, e.g., partial cells (probably important for the steep DS topography) and adjoint (probably not relevant here).

7. P.7, line 20 (and elsewhere) – "water immediately mixed once it became denser than the water below"- why?, is vertical mixing enhanced based on stability?, does this contradict the "implicit vertical diffusion" (constant?) mentioned befdore?.

8. P.9, line 25 and Fig. 12- it seems that mass is not conserved whereas in the non-hydrostatic model the entire lake is less dense than the hydrostatic case?, please explain how this is possible if the same surface BCs are used.

9. P.12- line 31- delete "xxx" (missing texts?). Lines 9-10- "the hydrostatic simulations are not suitable for simulating fine resolution. . ."- one should add, "unless more sophisticated vertical mixing scheme than used here is applied".

---

## Referee Comment (RC2) · Anonymous Referee #2 · 11 Jul 2017

Dear Editor, Please find my review on the paper "Non-hydrostatic effects in the Dead Sea", Padon and Ashkenazy, submitted to the OS.

The aim of this paper is to investigate the role of non-hydrostatic processes in the hypersaline Dead Sea through a series of numerical simulations using the MITgcm. The model is run on three computational grids with horizontal resolutions of 400, 200, and 100 m and surface forcing consisting of idealized annual and diurnal forcing through relaxation of surface temperature and salinity, and either constant or diurnally varying winds. The model is run in both hydrostatic and non-hydrostatic modes and the results compared. For the 400 and 200 m grid simulations the results are nearly identical. In the 100 m grid simulations the only significant difference occurred at night in the winter diurnal forcing simulations where the non-hydrostatic model maintained an unstable

stratification across the northern two-thirds of the lake for several hours before dawn with the upper water denser than the deep water by as much as 0.05 kg m-3. In the corresponding hydrostatic simulation this instability was restricted to a much smaller region with values of less than 0.001 kg m-3. These results may be an interesting curiosity but they appear to be very minor and restricted to very specific and limited situations and therefore it is doubtful that this non-hydrostatic phenomenon plays a significant role in determining the overall circulation in the Dead Sea. The manuscript contains several serious flaws. If the authors wish to convince the reader that these non-hydrostatic effects are responsible for maintaining unstable stratification for many hours they need to prove that such events really occur in this lake and then provide a more convincing explanation. In figure 11 the authors present data that is supposed to support the claim of existence of colder and denser surface water overlaid warmer water for hours, however the authors admit (in the figure caption) that the cooler surface water can be due to river runoff that keep the surface water diluted, cooler and less dens than the underlying brine. Thus it seems that fig 11 cannot be considered as supporting evidence, unless positive evidence of both salinity and temperature point that an unstable situation really exists. Figures 9 and 10 present the time series of the significant differences between the hydrostatic and non-hydrostatic simulations. The main difference is the instability and convective mixing of the dense plume in the early morning hours. On one hand the authors suggest that this is a feature that appears to be unique to the non-hydrostatic simulations. Yet in Figure 7 they indicate that it also occurs, albeit to a lesser extent, in the hydrostatic simulations. The location chosen to assess this effect is the center of the lake, however it would make more sense to compare the density difference time series at a point in the northeastern part of the lake where apparently the hydrostatic simulation also produces this instability. Regarding the experimental setup, in the 100 m runs the do not use the convective parameterization that was used in the 400 and 200 m runs, claiming that the 100 m resolution should be fine enough to explicitly simulate convective mixing. Furthermore in the 400 and 200 m grid simulations they use the simplest scheme available in the model (implicit vertical diffusion). One would expect including 100 m grid simulations that include the convective parametrization in order to be able to properly compare these results to the coarser grid results. It is unclear why switching off the convection scheme in the 100 m non-hydro run while leaving it turned on in the hydro run? This raises serious doubts about demonstrating conclusively that the effects they are seeing are physical or just an artifact of the convection scheme. The choice of the case of the Dead Sea for exploring the non-hydrostatic effect is not clear, the there are many narrow and deep lakes on earth, which experience winter convection driven by surface cooling. These freshwater lakes are much simpler to explore, without the added complexity of the Dead Sea. Anyway, a proper validation of the model results that unstable density structure can remain for the entire nighttime is missing in this paper. Minor commnet: Page 3, lines 20-25 talk about the complications and computational expense of running non-hydrostatic simulations. It is unclear what the message in this paragraph is? Line 30 - only 15 levels in the vertical are used, this is very coarse, with the upper five levels having thicknesses of 5 m each. While this may be ok for the winter when the stratification is weak to nonexistent, it seems to be very problematic in the summer.

Page 4, line 5: with an annual cycle in the forcing the model will never reach a steady state. It may reach a repeating annual cycle, which was not demonstrated.

Page 5, lines 16-17: on what basis do the authors pick the restoring time scales for the surface forcing of 12 and 4 days for S and T respectively? This is especially problematic for the diurnal cycle experiments. Also, the diurnal cycle experiments only use T forcings since S in constant.

Pgae 9: In their 100 m resolution runs, the hydrostatic simulations include the convective parameterization while the non-hydrostatic model has the parameterization turned off with the explanation that the 100 m resolution should be able to explicitly simulate the convective mixing. From a modeling perspective this is probably the major weakness in the manuscript and eliminates the possibility of attributing the differences in the result to non-hydrostatic effects. Line 12 – why the authors compare to Gulf of Eilat?

What is the relevance of such a comparison where the systems are so different? Lines 18-19 (and Fig 11): in the figure caption (fig 11) the authors say that the cooler water can be due to dilution, which means that freshwater input may be sufficient to neutralize the effect of cooling in terms of density. If that is the case, then the simulations based on T forcing alone may be a curiosity but they have no real significance or value for the Dead Sea. Regarding the summer simulations, it is not clear that an upper model layer thickness of 5 m is adequate to simulate the shallow summertime convection, which is primarily wind forced with possibly some help from night time cooling. In table 1 it seems that the day-night temp difference for forcing was chosen to be 10 deg, this seems very unrealistic.
* * *

---

## Author Comment (AC1) · 15 Aug 2017

We thank the first referee for his deep and thoughtful report. The referee identified himself as Tal Ezer, an expert on ocean modeling who developed and run the first oceanographic model to study the water circulation in the Dead Sea. We are grateful for his report.

Numerical modeling of the circulation and mixing processes in the extreme environment of the Dead Sea (DS) is a very challenging task, so progress has been very limited since the development of the first coarse-resolution circulation models of this lake more than three decades ago (Ezer, 1984; Sirkes, 1986). Therefore, the current study, using advanced high-resolution, non-hydrostatic MITgcm, provides new insights

into the dynamics of the DS, with focus on non-hydrostatic processes. Because of the unique characteristics of the DS, adapting to the DS model schemes and parameterizations derived for other oceans is usually not appropriate. Therefore, even this advanced model lacks some realism in aspects such as the equation of state, wind drag coefficient, lack of freshwater input and possibly unrealistic mixing schemes. Nevertheless, in my opinion, in the context of model sensitivity experiments to compare hydrostatic versus non-hydrostatic dynamics, the study is important and novel enough to be published, if these limitations are acknowledged. Since the focus of this study is on the surface mixing process, clarifications are needed about the (somewhat unrealistic) model mixing parameters and imposed surface boundary conditions. It seems that the main difference between the hydrostatic and non-hydrostatic simulations during winter nights is a direct result of the different vertical mixing used in those cases and the question is whether this is an artifact of the particular mixing used by the model or a more general result. The paper is generally quite well written (though some text and figures can be improved; see comments below) and the interesting results are clearly presented.

We thank the referee for his careful and accurate summary of our study. See our detailed response below.

Major comments:

1. The main concern is the vertical mixing used, which needs further clarifications (p. 4). In the hydrostatic model runs, slight instability due to surface cooling caused immediate strong mixing- this is typical for mixing that depends say on the Richardson number or based on a stability function in a turbulence scheme like the Mellor-Yamada (M-Y) model, but here it seems that (inappropriate) constant vertical diffusion coefficient is used everywhere, which do not explain the results. Line 15 is confusing, since only Large et al. (1994) describe the KPP, while Mellor and Yamada (1982) and Ezer (2005) describe and use the M-Y turbulence scheme, not KPP. The imposed restoring surface BCs (p. 5) can really limits surface variability and strongly impact the mixed

layer variability (especially for hydrostatic models), so one wonders how different the results would have been without this restriction. Also, fresh water input from the Jordan river and winter flash floods which are neglected here may play a major role in the seasonal changes in stratification – this is only acknowledged at the end of the paper, but this limitation on the salinity and density fields should be clearly indicated when the model setup and surface BCs are described.

As mention on page 4 (line 15), we have used both the implicit vertical diffusion and the KPP vertical mixing schemes of MITgcm and both yielded similar results. In the implicit vertical diffusion scheme, a much larger vertical diffusion coefficient is set when unstable conditions are identified–this check is performed for each grid point at each time step and is aimed to parameterize the convection process. We clarified this point on page 4 as follows:

> In all hydrostatic simulations and the low resolution non-hydrostatic simulations (400 m and 200 m), implicit vertical diffusion was used. The implicit vertical diffusion scheme is a standard vertical mixing scheme (MITgcm-group, 2010) in which the vertical diffusion is drastically increased (i.e., from a value of $10\text{-}5$ m$^2$ s-1 15 to 10 m$^2$ s-1) when the water column, at each time step and each grid point, becomes unstable.

As for the inaccurate citation of Mellor and Yamada (1982); Ezer (2005), we now exclude these references.

The restoring times we used for the temperature and salinity are close to the values used by the global ocean simulation of MITgcm (i.e., restoring times for temperature and salinity of 2 and 6 months where the top ocean layer depth is 50 m) and to the values discussed in Tziperman et al. (1994); the depth of top layer in our simulations is 5 m and hence the approximately one order of magnitude smaller restoring times of 4 and 12 days. These restoring times yielded sea surface temperature and salinity that are close to the observed annual cycle of surface temperature and salinity. We indeed

aware that our results may be different when using drastically different restoring times and we acknowledge this in the revised manuscript (in the paragraph after Eq. 3) as follows:

> ... were $12$ days for surface salinity and $4$ days for surface temperature; we note that a significantly different choice of restoring times may yield different results.

In the revised manuscript we now mention in the model setup section that we ignore the Jordan river inflow and winter flash floods (page 5):

> The influx of fresh water from the Jordan river and winter flash floods are ignored here and indirectly considered through the relaxation to fresher sea surface during the winter.

2. Though this is a numerical modeling study, it would be useful to add (if possible, given the limited availability) some direct comparison with observation, to show at least that the seasonal variations in temperature profiles are reasonable (measurements by Anati, Hecht, Gertman and others are cited, and a 1997's book on the DS has further data). The only showing of some observations (Fig. 11) is difficult to relate to any model shown results (is it possible to show example of similar model results?).

Following the referee recommendation we now include in the revised manuscript data of temperature and salinity profiles reconstructed from the IOLR web-page (Fig. 5 of the revised manuscript). These are similar to the simulations results shown in Fig. 4 of the revised manuscript. This figure is also shown here (Fig. 1 below). In addition, Figs. 10, 11 of the revised manuscript basically show similar situation as in the observation (which are now shown in Fig. 12).

Other comments:

3. While the paper is quite well written, the text can be improved- some examples: -
The second sentence of the abstract is awkwardly written - the water level of the DS
"has been dropping" for a long time, it is not a new phenomenon, as can be interpreted
here. - P. 2, line 14, "denser" instead of "heavier". - P. 10, title- "Diurnal Cycle Under
Fixed Diurnally Varying Winds" is a confusing statement (is it fixed or varying?), what
is probably meant is spatially even but temporally varying. And same page line 5- what
is meant by "climatological means"? hourly, daily, monthly means?

Following the referee comments, we improved the second sentence of the abstract as
follows:

> For at least three decades, the Dead Sea's water level has been dropping
> by more than 1 m per year, . . .

In addition, we have replaced the word "heavier" by "denser". We changed the title of
section 5 to ". . . Diurnal Cycle Under Spatially Fixed Diurnally Varying Winds". As for
the "climatological means" mentioned at the same page, as mentioned in Sec. 2.2,
we refer here to the diurnal mean cycle (based on hourly mean data) averaged over
January and July over years 2006-2010. To avoid confusion we refer the reader to
Section 2.2 and Fig. 2 and now write "climatological hourly means".

4. Figures can be improved with clarifications in captions- some suggestions: - Fig. 1-
since it is based on data from 1978 when water level was much higher, it should be
clarified for what year the shown shoreline is and what is the water level at that year. -
Fig. 2- may be show wind vectors instead of components?, also caption should state if
these data are based on hourly observations and what source. - Fig. 4- may be show
only the top 100 m (like other figures) to see the details.

We agree and corrected mentioned captions and figures as follows. The bathymetry
shown in Fig. 1 corresponds to surface water level of -427 meter below mean sea
level (year 2013)–the first sentence of Fig. 1 caption is now: "Dead Sea bathymetry

(based on Hall, 1978), corresponds to surface water level of -427 meter below mean sea level (year 2013).". As for Fig. 2, we prefer to leave this figure as is, to allow better visualization of the fine details of the different wind stress component during the different months. We now write at the end of caption of Fig. 2 that

> The curves are based on hourly mean wind data of the Israel Meteorological Service's Ein Gedi station.

We have changed Fig. 4 as suggested by the referee–now we show only the top 100 m.

5. P.2, near bottom, to first paragraph of p.3- "non-hydrostatic effects are expected to be significant"- is it possible to quantify this assumption based on say scaling arguments such as ratio of horizontal and vertical velocity scales in the DS vs open oceans and the very steep bottom slopes of the DS?. Also, when citing past studies of hydrostatic vs. non-hydrostatic comparisons, one should add that these studies may not be applicable to the unique environment of the DS. For example, in the DS the very high salinity and variations in salinity (which are ignored here) play a major role in the static stability compared to the major role of temperature in most other oceans.

Indeed it is possible to quantify the non-hydrostatic effects based on the non-dimensional number (nonhydrostatic parameter) $n$ developed by Marshall et al. (1997):

$$n = \frac{\gamma^2}{R_i} = \frac{U^2}{L^2 N^2}, \tag{1}$$

where $\gamma = H/L$ ($H$ and $L$ are the vertical and horizontal scales), and $R_i$ is the Richardson number, $R_i = N^2 H^2/U^2$, where $N$ is the buoyancy (Brunt-Väisälä) frequency, $N = -(g/\rho_0)(\partial\rho/\partial z)$, and $U$ is the horizontal velocity scale. The hydrostatic approximation holds when $n \ll 1$. During the summer, $N \approx 8 \times 10^{-3}$ s$^{-1}$ and with $U \approx 0.1$ m s$^{-1}$, $H \approx 300$ m, $L \approx 20$ km, yielding $n \approx 4 \times 10^{-7} \ll 1$. Thus, the hydrostatic relation

holds during the summer. The buoyancy frequency becomes much smaller during the winter and approaches zero when the entire water column mixes; $n$ is expected to be much larger than 1 then. When the bathymetry is steep (relatively large $\gamma$), the water column becomes nonhydrostatic even faster.

We find the above discussion too complicated to actually improve the understanding of the nonhydrostatic effect in the context of the Dead Sea. In essence, the most important ingredient is the stratification–as the water column becomes weakly stratified, nonhydrostatic effects are expected to be significant. In the revised manuscript we refer the interested reader to Eq. (2) of Marshall et al. (1997).

Following the referee comments, we also included the following sentence (top of page 3 of the revised manuscript):

> One should note however that, due to the unique environment of the Dead Sea, these previous studies may not be applicable to the Dead Sea; e.g., salinity variations affect significantly the stability of the water column in the Dead Sea in contrast to most of the world oceans in which salinity variations play less significant role (in comparison to the temperature variations).

6. P.3- in the description of the model, it is not clear if some of the features are actually used here, e.g., partial cells (probably important for the steep DS topography) and adjoint (probably not relevant here).

Correct, the partial cells option is used but the adjoint. We clarify this in the revised manuscript (section 2.1 of the revised manuscript) as follows:

> In the current study we use the hydrostatic and full non-hydrostatic options of MITgcm together with the partial cells option, to account for the steep bathymetry of the Dead Sea.

7. P.7, line 20 (and elsewhere) – "water immediately mixed once it became denser

than the water below"- why?, is vertical mixing enhanced based on stability?, does this contradict the "implicit vertical diffusion" (constant?) mentioned before?.

We used the wrong terminology. We now replace the words "immediately mixed" by "very fast mixed" (according to the time associated with the implicit vertical diffusion coefficient).

8. P.9, line 25 and Fig. 12- it seems that mass is not conserved whereas in the non-hydrostatic model the entire lake is less dense than the hydrostatic case?, please explain how this is possible if the same surface BCs are used.

One should note that in the winter (January) simulation we did not run the model to a steady state as the winter is a transient phenomenon (in which, e.g., there is a positive net freshwater flux and surface cooling). During this relatively short simulation one may achieve different mean density between the hydrostatic and nonhydrostatic simulations, due to the different dynamics of the two.

9. P.12- line 31- delete "xxx" (missing texts?). Lines 9-10- "the hydrostatic simulations are not suitable for simulating fine resolution. . ."- one should add, "unless more sophisticated vertical mixing scheme than used here is applied".

We deleted the extra "xxx" and added the end of the sentence suggested by the referee.

**References**

Anati, D. A. (1999). The salinity of hyper saline brines: Concepts and misconceptions. *Int. J. Salt Lake Res.*, 8:55–70.

Ezer, T. (2005). Entrainment, diapycnal mixing and transport in three-dimensional bottom gravity current simulations using the mellor–yamada turbulence scheme. *Ocean Modelling*, 9(2):151–168.

Gertman, I. and Hecht, A. (2002). The Dead Sea hydrography from 1992 to 2000. *J. Mar. Sys.*, 35(3-4):169–181.

Hall, J. K. (1978). Dead Sea Geophysical Survey, Bathymetric Chart. Marine Geology Division, Geological Survey of Israel.

Large, W. G., Mcwilliams, J. C., and Doney, S. C. (1994). Oceanic vertical mixing: A review and a model with a nonlocal boundary-layer parameterization. *Rev. Geophys.*, 32(4):363–403.

Marshall, J., Hill, C., Perelman, L., and Adcroft, A. (1997). Hydrostatic, quasi-hydrostatic, and nonhydrostatic ocean modeling. *J. Geophys. Res.*, 102(C3):5733–5752.

Mellor, G. L. and Yamada, T. (1982). Development of turbulence closure model for geophysical fluid problems. *Rev. Geophys. Space Phys.*, 20(4):851–875.

MITgcm-group (2010). MITgcm User Manual. Online documentation, MIT/EAPS, Cambridge, MA 02139, USA. http://mitgcm.org/public/r2_manual/latest/online_documents/manual.html.

Tziperman, E., Toggweiler, J. R., Feliks, Y., and Bryan, K. (1994). Instability of the thermohaline circulation with respect to mixed boundary-conditions: Is it really a problem for realistic models. *J. Phys. Oceanogr.*, 24(2):217–232.
* * *
[Figure]

**Fig. 1.** Salinity (a) and temperature (b) profiles measured by the IOLR at the deepest point of the Dead Sea. The simulations shown in Fig. 4 are in the agreement with these observations.

---

## Author Comment (AC2) · 15 Aug 2017

We thank Referee 2 for the helpful comments on the submitted manuscript. Please see below our detailed response.

The aim of this paper is to investigate the role of non-hydrostatic processes in the hypersaline Dead Sea through a series of numerical simulations using the MITgcm. The model is run on three computational grids with horizontal resolutions of 400, 200, and 100 m and surface forcing consisting of idealized annual and diurnal forcing through relaxation of surface temperature and salinity, and either constant or diurnally varying winds. The model is run in both hydrostatic and non-hydrostatic modes and the results compared. For the 400 and 200 m grid simulations the results are nearly identical. In

[Figure]

the 100 m grid simulations the only significant difference occurred at night in the winter diurnal forcing simulations where the non-hydrostatic model maintained an unstable stratification across the northern two-thirds of the lake for several hours before dawn with the upper water denser than the deep water by as much as 0.05 kg m-3. In the corresponding hydrostatic simulation this instability was restricted to a much smaller region with values of less than 0.001 kg m$^{-3}$.

We thank the referee for the accurate summary of our study.

These results may be an interesting curiosity but they appear to be very minor and restricted to very specific and limited situations and therefore it is doubtful that this non-hydrostatic phenomenon plays a significant role in determining the overall circulation in the Dead Sea.

Our simulations indicate that nonhydrostatic effects affect significantly the overall circulation of the Dead Sea during winter. Following previous studies (e.g., Marshall et al., 1997) nonhydrostatic effects should be taken into account under weak stratification conditions and when the ratio between depth and length scales is relatively large, as in the Dead Sea. Yet since our setup is idealized (mainly due to the idealized surface forcing), it is possible that the nonhydrostatic effect will be less significant under more realistic forcing. We also note even small changes in the Dead Sea due to nonhydrostatic effects may be important for the potash industry of the Dead Sea.

The manuscript contains several serious flaws. If the authors wish to convince the reader that these non-hydrostatic effects are responsible for maintaining unstable stratification for many hours they need to prove that such events really occur in this lake and then provide a more convincing explanation. In figure 11 the authors present data that is supposed to support the claim of existence of colder and denser surface water overlaid warmer water for hours, however the authors admit (in the figure caption) that the cooler surface water can be due to river runoff that keep the surface water diluted, cooler and less dens than the underlying brine. Thus it seems that fig 11 cannot be

considered as supporting evidence, unless positive evidence of both salinity and temperature point that an unstable situation really exists.

Unfortunately there are no continuous measurements of salinity as for temperature–this is since standard conductivity (salinity) device is not suitable for the Dead Sea water and densimeter is needed to measure the density and quasi-salinity. Every several month the IOLR is taking hydrographic measurements at the deepest point of the Dead Sea (some of which are shown in new Fig. 5 of the revised manuscript) but these, unfortunately, cannot support the main claim of the paper, as they reflect the state of the water column in a given location at a certain time. Although we admit that floods of freshwater may stand behind the observed colder water overlaying warmer water (shown in Fig. 12 of the revised manuscript), one has to remember that such floods occur several times during winter and contribute significantly to the lower surface salinity values during winter. The sea surface in the model is restored to these lower salinity values, and thus reflect the mean effect of freshwater floods and precipitation. Thus, we believe that our simulations account for, indirectly, the effect of flooding, and thus should be trusted.

Figures 9 and 10 present the time series of the significant differences between the hydrostatic and non-hydrostatic simulations. The main difference is the instability and convective mixing of the dense plume in the early morning hours. On one hand the authors suggest that this is a feature that appears to be unique to the non-hydrostatic simulations. Yet in Figure 7 they indicate that it also occurs, albeit to a lesser extent, in the hydrostatic simulations.

To our understanding, convective water plumes cannot be generated under the hydrostatic assumption. The convection parameterization schemes (like convective adjustment, implicit vertical diffusion and KPP), basically only mix rapidly the vertically unstable cells. The unstable situation we describe for the hydrostatic case is very short compare to the plumes' time scale–it is associated with the increased diffusion coefficient of either the implicit vertical diffusion coefficient or the KPP. Thus, although

possible, as demonstrated in Figs. 10 and 11 of the new manuscript (old Figs. 9 and 10), unstable water column is very limited both in space (Fig. 7) and time compare to the nonhydrostatic simulations.

The location chosen to assess this effect is the center of the lake, however it would make more sense to compare the density difference time series at a point in the northeastern part of the lake where apparently the hydrostatic simulation also produces this instability.

Exactly for this reason we show a cross section in Fig. 9 of the revised manuscript (old Fig. 8) that clearly show that the hydrostatic simulation is very different than the nonhydrostatic across the entire lake. Below (Figs. 1, 2) we present Figs. 10 and 11 of the revised manuscript (old Figs. 9, 10) for a point located at the northeastern part of the lake ($X = 15$ km, $Y = 40$ km), close to the eastern coast. These two figures resemble closely the results we obtain for the middle of the lake, shown in Figs. 10, 11 of the revised manuscript.

Regarding the experimental setup, in the 100 m runs the do not use the convective parameterization that was used in the 400 and 200 m runs, claiming that the 100 m resolution should be fine enough to explicitly simulate convective mixing. Furthermore in the 400 and 200 m grid simulations they use the simplest scheme available in the model (implicit vertical diffusion). One would expect including 100 m grid simulations that include the convective parametrization in order to be able to properly compare these results to the coarser grid results. It is unclear why switching off the convection scheme in the 100 m non-hydro run while leaving it turned on in the hydro run? This raises serious doubts about demonstrating conclusively that the effects they are seeing are physical or just an artifact of the convection scheme.

Our main simulations are the 100 m resolution and the coarser resolution runs are mainly aimed to generate initial conditions for the fine, 100 m, simulations. Following Marshall et al. (1997) and experts that we consulted with (one of the developers of MIT-

gcm and another world expert oceanographer), one should not use convection param- eterization when the convection process is simulated (through the vertical momentum equation). In the absence of the full vertical momentum equation, the convection pro- cess cannot fully simulated and hence the convective parametrization of the hydrostatic simulations. Known examples of MITgcm of convection processes, like deep convec- tion and plume on the slope do not include convection schemes. We also note that we the implicit vertical diffusion scheme is not the simplest convection scheme (the con- vective adjustment scheme is simpler) and that we have obtained similar results when using the more complicated KPP scheme (Large et al., 1994).

The choice of the case of the Dead Sea for exploring the non-hydrostatic effect is not clear, the there are many narrow and deep lakes on earth, which experience winter convection driven by surface cooling. These freshwater lakes are much simpler to explore, without the added complexity of the Dead Sea.

We agree that there are better lakes to study nonhydrostatic effects. Yet, our main goal is to study the Dead Sea circulation, and nonhydrostatic effects within this unique lake. Dead Sea circulation is of local and regional importance, and we are the first to study nonhydrostatic effects in this lake.

Anyway, a proper validation of the model results that unstable density structure can remain for the entire nighttime is missing in this paper.

As we elaborated above, this is indeed a limitation of our study and thus the results presented in the paper should be regarded as model predictions. We hope that con- tinuous salinity measurements will be available in the future, and base on these we hopefully will be able to validate these model predictions.

Minor comment:

Page 3, lines 20-25 talk about the complications and computational expense of running non-hydrostatic simulations. It is unclear what the message in this paragraph is?

Following the referee comment, we deleted this paragraph.

Line 30 - only 15 levels in the vertical are used, this is very coarse, with the upper five levels having thicknesses of 5 m each. While this may be ok for the winter when the stratification is weak to nonexistent, it seems to be very problematic in the summer.

As mentioned in the revised manuscript (Sec. 2.2), we have repeated some of the numerical simulations (winter time) using 100 vertical levels (instead of 15), with upper ocean vertical resolution of 1 m and obtain similar results. As for the summer time, we have looked at temperature measurements from the center of the Dead Sea and it is apparent that the nocturnal cooling span more than the upper 10 m, such that our model could resolve this cooling.

Page 4, line 5: with an annual cycle in the forcing the model will never reach a steady state. It may reach a repeating annual cycle, which was not demonstrated.

Yes, we meant "quasi repeating annual cycle" and we change the text accordingly as follows (end of the first paragraph of Section 2.2):

> it was run for twenty years to a quasi repeating annual cycle, representing the Dead Sea's circulation under annual cycle forcing.

This quasi repeating annual cycle results are demonstrated in Fig. 4 of the revised manuscript.

Page 5, lines 16-17: on what basis do the authors pick the restoring time scales for the surface forcing of 12 and 4 days for S and T respectively? This is especially problematic for the diurnal cycle experiments. Also, the diurnal cycle experiments only use T forcings since S in constant.

We pick the restoring times based on Tziperman et al. (1994) and on the global ocean example of MITgcm for which the restoring time for temperature is three time smaller than the restoring time for salinity—for a top layer depth of 50 m the temperature and

salinity restoring times are 2 and 6 months. The depth of the top layer in our setup is 5 m and thus restoring times of 4 and 12 days for temperature and salinity are reasonable. In addition, these restoring times yielded reasonable agreement with observations (see Fig. 5 of the revised manuscript). As now mentioned in the revised manuscript (in the paragraph after Eqs. 2,3), we agree that different restoring times may yield different results:

> . . . were $12$ days for surface salinity and $4$ days for surface temperature; we note that a significantly different choice of restoring times may yield different results.

Pgae 9: In their 100 m resolution runs, the hydrostatic simulations include the convective parameterization while the non-hydrostatic model has the parameterization turned off with the explanation that the 100 m resolution should be able to explicitly simulate the convective mixing. From a modeling perspective this is probably the major weakness in the manuscript and eliminates the possibility of attributing the differences in the result to non-hydrostatic effects.

Please see our response above to this comment.

Line 12 – why the authors compare to Gulf of Eilat? What is the relevance of such a comparison where the systems are so different?

We mention the Gulf of Eilat to demonstrate that indeed dense water can overlay lighter water for several hours as we observe in our model. The gulf of Eilat has similar surface temperature forcing as the Dead Sea and similar precipitation rate.

Lines 18-19 (and Fig 11): in the figure caption (fig 11) the authors say that the cooler water can be due to dilution, which means that freshwater input may be sufficient to neutralize the effect of cooling in terms of density. If that is the case, then the simulations based on T forcing alone may be a curiosity but they have no real significance or value for the Dead Sea.

See our response above. We agree and clearly mention this limitation in the text. Please note that freshwater water floods occur in the Dead Sea and contribute to the fresher sea surface during winter. Thus, our restoring to the fresher winter sea surface indirectly (and in an average way) take into account such flooding events. We hope that in the future continuous salinity measurements will be available to validate (or reject) our model predictions.

Regarding the summer simulations, it is not clear that an upper model layer thickness of 5 m is adequate to simulate the shallow summertime convection, which is primarily wind forced with possibly some help from night time cooling.

See above our response on this point.

In table 1 it seems that the day-night temp difference for forcing was chosen to be 10 deg, this seems very unrealistic.

Please note that the restoring times in these simulations are much larger than 1 day (4 and 12 days for temperature and salinity) such that the actual forcing does not reach a day-night difference of 10C.

**References**

Anati, D. A. (1999). The salinity of hyper saline brines: Concepts and misconceptions. *Int. J. Salt Lake Res.*, 8:55–70.

Ezer, T. (2005). Entrainment, diapycnal mixing and transport in three-dimensional bottom gravity current simulations using the mellor–yamada turbulence scheme. *Ocean Modelling*, 9(2):151–168.

Gertman, I. and Hecht, A. (2002). The Dead Sea hydrography from 1992 to 2000. *J. Mar. Sys.*, 35(3-4):169–181.

Hall, J. K. (1978). Dead Sea Geophysical Survey, Bathymetric Chart. Marine Geology Division, Geological Survey of Israel.

Large, W. G., Mcwilliams, J. C., and Doney, S. C. (1994). Oceanic vertical mixing: A review and a model with a nonlocal boundary-layer parameterization. *Rev. Geophys.*, 32(4):363–403.

Marshall, J., Hill, C., Perelman, L., and Adcroft, A. (1997). Hydrostatic, quasi-hydrostatic, and nonhydrostatic ocean modeling. *J. Geophys. Res.*, 102(C3):5733–5752.

Mellor, G. L. and Yamada, T. (1982). Development of turbulence closure model for geophysical fluid problems. *Rev. Geophys. Space Phys.*, 20(4):851–875.

MITgcm-group (2010). MITgcm User Manual. Online documentation, MIT/EAPS, Cambridge, MA 02139, USA. http://mitgcm.org/public/r2_manual/latest/online_documents/manual.html.

Tziperman, E., Toggweiler, J. R., Feliks, Y., and Bryan, K. (1994). Instability of the thermohaline circulation with respect to mixed boundary-conditions: Is it really a problem for realistic models. *J. Phys. Oceanogr.*, 24(2):217–232.

[Figure]

**Fig. 1.** Density profile versus time in the northeastern part of the lake (X=15 km, Y=40km), for the hydrostatic and the non-hydrostatic simulations of the diurnal cycle with a 100m resolution, in winter (Jan)

[Figure]

**Fig. 2.** Winter density difference between surface and deep water at a northeastern point (X=15 km, Y=40 km) versus time, for the hydrostatic and the non-hydrostatatic runs of the diurnal cycle (100 m).